# Outpacing conventional nicotinamide hydrogenation catalysis by a strongly communicating heterodinuclear photocatalyst

Linda Zedler [1,2,5], Pascal Wintergerst[3,5], Alexander K. Mengele [3], Carolin Müller [1,2], Chunyu Li[1,2], Benjamin Dietzek-Ivanšić [1,2,4 ✉] & Sven Rau [3 ✉]

Unequivocal assignment of rate-limiting steps in supramolecular photocatalysts is of utmost importance to rationally optimize photocatalytic activity. By spectroscopic and catalytic analysis of a series of three structurally similar $[(tbbpy)_2Ru-BL-Rh(Cp^*)Cl]^{3+}$ photocatalysts just differing in the central part (alkynyl, triazole or phenazine) of the bridging ligand (BL) we are able to derive design strategies for improved photocatalytic activity of this class of compounds (tbbpy = 4,4´-*tert*-butyl-2,2´-bipyridine, $Cp^*$ = pentamethylcyclopentadienyl). Most importantly, not the rate of the transfer of the first electron towards the $Rh^{III}$ center but rather the rate at which a two-fold reduced $Rh^I$ species is generated can directly be correlated with the observed photocatalytic formation of NADH from $NAD^+$. Interestingly, the complex which exhibits the fastest intramolecular electron transfer kinetics for the first electron is not the one that allows the fastest photocatalysis. With the photocatalytically most efficient alkynyl linked system, it is even possible to overcome the rate of thermal NADH formation by avoiding the rate-determining β-hydride elimination step. Moreover, for this photocatalyst loss of the alkynyl functionality under photocatalytic conditions is identified as an important deactivation pathway.

[1] Institute of Physical Chemistry, Friedrich Schiller University Jena, Helmholtzweg 4, 07743 Jena, Germany. [2] Leibniz Institute of Photonic Technology Jena, Department Functional Interfaces, Albert-Einstein-Straße 9, 07745 Jena, Germany. [3] Institute of Inorganic Chemistry I, Materials and Catalysis, Ulm University, Albert-Einstein-Allee 11, 89081 Ulm, Germany. [4] Center for Energy and Environmental Chemistry Jena (CEEC Jena), Philosophenweg 7a, 07743 Jena, Germany. [5] These authors contributed equally: Linda Zedler, Pascal Wintergerst. ✉email: benjamin.dietzek@uni-jena.de; sven.rau@uni-ulm.de

One of the biggest challenges of the century is to turn the currently unsustainable energy production upside down[1]. A promising approach to deal with the inherent inter-mittencies of renewable energies is to store them in chemical bonds[2]. Photocatalytic water splitting using molecular compo-nents could represent one of many viable solutions to this goal[2,3]. In this context, di- or oligonuclear supramolecular photocatalysts consisting of a chromophoric moiety and a catalytic center con-nected by an electron transporting bridging ligand (BL) might represent a very promising approach[4].

Much work has been performed to understand the molecular prerequisites for optimized photocatalytic output using supra-molecular photocatalysts[5,6]. As the organic BL is at the heart of such oligonuclear photocatalysts, it is essential to fine tune this component in order to allow for multiple unidirectional elec-tron transfers between chromophore and catalyst[7]. Thus, a plethora of synthetic optimization work, supported by (time-resolved) spectroscopy has been done to understand the cor-relation of structural changes of the BL and the photocatalytic output measured in terms of turnover number (TON) or turnover frequency (TOF)[6,8–10]. Examples for such rational synthetic approaches, are the attempts to stabilize excited state properties by introducing electron-withdrawing substituents in the BL by Karnahl et al., which however led to a decrease in electron transfer rate and in turn lowered catalytic activity[8]. Also Ishitani et al. could show that the tuning of the electronic properties of the BL requires careful optimization in order to balance the rate of intramolecular electron transfer and turnover at the catalytic center[11].

However, the complexity of the investigated light-driven multielectronic processes, quite often renders the assignment of the bottleneck in supramolecular photocatalysts difficult, i.e., identifying the one most critical parameter to lift limitations in the photocatalytic activity is—generally speaking—an unsolved problem. In many cases it is not clear whether e.g., the sum of

various visible light-driven electron transfer processes to activate the catalyst or turnover at the photochemically activated catalyst represented the bottleneck. Some insights are only available for intermolecular light-driven water oxidations[12,13].

Nevertheless, as light-induced electron transfer within supramolecular catalysts can proceed within the ultrafast time domain[5,14], such catalysts hold the potential to be more efficient than thermally operating catalysts. Provided suitable BL architectures can be identified that allow rapid multiple charge accumulation at the catalytic center, the pho-tochemical scenario might outpace the ground-state reactivity of the systems which can be hampered by high activation barriers. In order to investigate this enticing hypothesis a set of reference catalysts is required which are capable of per-forming the same catalytic reaction in a photochemical and thermal setting. Furthermore, the systems have to allow for the separation of the kinetics of light-driven redox activation of the catalytic center from actual catalytic turnover to identify the overall rate-limiting step.

A very promising example for this purpose are [(NN)Rh(Cp*) Cl]-based catalysts (NN = α-diimine ligand, Cp* = pentamethyl-cyclopentadienyl) which are well understood in terms of their redox chemistry[15–17] and the mechanism of photochemical or thermal, formate-driven selective nicotinamide reduction[18–20]. Hence, such [(NN)Rh(Cp*)Cl]-based catalysts have found wide-spread applications in the area of cofactor recycling for enzymatic reactions[21–24]. As shown in Fig. 1a on the left, only after a two-fold photochemical reduction of the Rh$^{III}$ center, the light-independent hydride transfer onto the nicotinamide derivative takes place, initiated by proton addition to the photochemically generated Rh$^I$ species[15].

Furthermore, these [(NN)Rh(Cp*)Cl]-based systems feature the possibility that both photochemical formation of Rh$^I$ [14] and catalytic turnover[18,25] can be evaluated independently from each other (see Fig. 1a). However, the reactivity of the

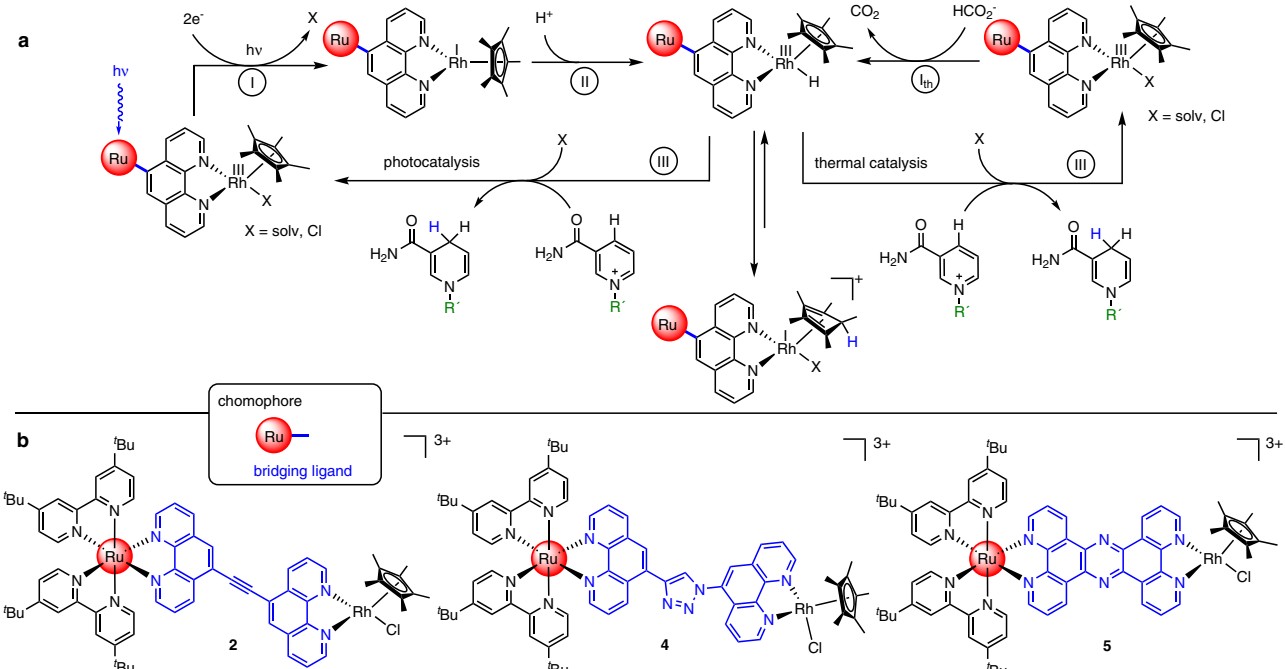

**Fig. 1 Mechanisms and investigated compounds for nicotinamide reduction. a** Depiction of the photocatalytic and thermal catalytic hydrogenation of nicotinamides by the selected class of rhodium catalysts[18–20,51]. Photocatalysis: (I) light-driven two electron reduction of Rh$^{III}$, (II) oxidative addition of a proton, (III) hydride transfer; Thermal catalysis: (I$_{th}$) coordination of formate and elimination of CO$_2$, (III) hydride transfer. **b** Molecular structure of the investigated photocatalysts **2**, **4** and **5**.

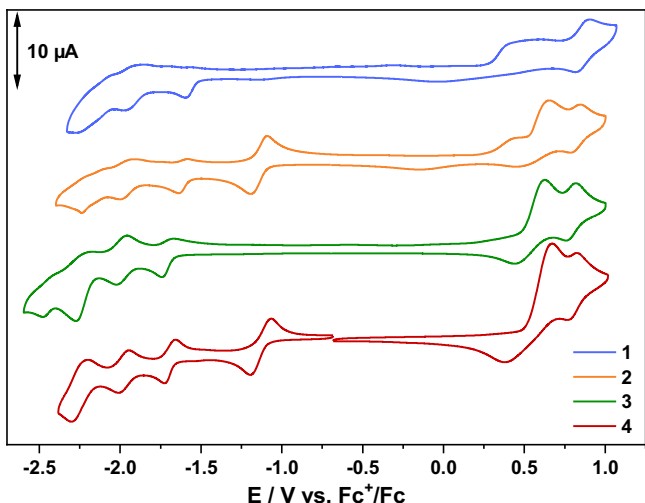

**Fig. 2 Synthesis route of complexes 1, 2, 3 and 4.** (i) (1) 5-ethynyl-1,10-phenanthroline, THF, diisopropylamine, [Pd(PPh$_3$)$_4$], CuI, 60 °C, 24 h; (2) KCN, dichloromethane/H$_2$O, rt, 1 h; (ii) (1) 5-azido-1,10-phenanthroline, CuSO$_4$, sodium ascorbate, H$_2$O/dichloromethane, rt, 24 h; (2) KCN, dichloromethane/H$_2$O, rt, 1 h; (iii) [Rh(Cp*)Cl$_2$]$_2$, dichloromethane, rt, 24 h.

**Fig. 3 Redox properties of the complexes.** Cyclic voltammograms of separate 1 mM acetonitrile solutions of **1** (blue), **2** (orange), **3** (green) and **4** (red) at room temperature with nBu$_4$NPF$_6$ as supporting electrolyte (0.1 M). An Ag wire is used as quasi reference electrode, Pt wire as the counter electrode and glassy carbon as the working electrode. All data referenced against Fc$^+$/Fc; scan rate = 100 mV s$^{-1}$.

[(NN)Rh(Cp*)Cl]-based systems heavily depends on the direct coordination environment, i.e., the choice of the NN-ligand[16].

In this work we therefore designed two structurally related photocatalysts utilizing a modular composition of the BLs. The two systems feature 1,10-phenanthroline (phen) coordination spheres on both the (tbbpy)$_2$Ru chromophore moiety as well as the RhCp* catalyst subunit and are either linked via an alkyne (complex **2**) or a triazole linker (complex **4**) in the easy to modify 5-position (see Fig. 1b)[26–28]. In combination with the recently reported tpphz (tetrapyridophenazine) derivative (complex **5**)[14,24], this allowed us to clearly assign for the first time photocatalytic reactivity changes in [(NN)$_2$Ru-BL-Rh(Cp*)Cl]$^{3+}$ systems to the different photophysics and photochemistries in these molecules caused by altered molecular structures of the BL.

## Results

**Synthesis**. Both new catalysts were prepared via chemistry on the complex (Fig. 2). The starting complexes for catalysts **2** and **4** were [(tbbpy)$_2$Ru(5-Br-phen)](PF$_6$)$_2$ and [(tbbpy)$_2$Ru(5-ethynyl-phen)](PF$_6$)$_2$ available from previous work, respectively (5-Br-phen = 5-bromo-1,10-phenanthroline, 5-ethynyl-phen = 5-ethy-nyl-1,10-phenanthroline)[26,27]. To access complex **1** we decided to construct the BL directly on the complex, although the ligand itself has been prepared before[29]. The bromo derivative was subjected to Sonogashira reaction on the complex with 5-ethinyl-phen in 55% yield[30]. Complex **3** was accessible via click reaction using the ethynyl substituted complex and 5-azido-1,10-phenanthroline[31] in 63% yield. For introduction of the rhodium center, complexes **1** and **3** were combined with [Rh(Cp*)Cl$_2$]$_2$ in methylene chloride to yield the corresponding target complexes **2** and **4** in nearly quantitative yields. Complex **5** was available from previous studies[24]. All complexes were characterized via NMR spectroscopy and high-resolution mass spectrometry (Supplementary Figs. 1–8).

**Redox properties**. The complexes were dissolved in acetonitrile containing $n$-Bu$_4$NPF$_6$ and investigated by cyclic voltammetry (Fig. 3). The Rh containing complexes show a reversible reduction at −1.16 (**2**) and −1.13 V (**4**) vs. Fc$^+$/Fc due to a two electron reduction of the Rh$^{III}$/Rh$^I$ couple[32,33]. At more negative potentials, i.e., between −1.6 and −2.2 V vs. Fc$^+$/Fc, **1**–**4** feature three reduction waves. At −1.9 V and −2.2 V vs. Fc$^+$/Fc the terminal bipyridine ligands become reduced[34,35], while the first reduction in the mononuclear complexes is localized on the BL. For the alkynyl complexes **1** and **2**, this reduction (−1.6 V vs. Fc$^+$/Fc) is shifted anodically by ~100 mV compared to the triazole complexes **3** and **4** at −1.7 V vs. Fc$^+$/Fc. All complexes feature a reversible Ru$^{III}$/Ru$^{II}$ oxidation at around 0.8 V vs. Fc+/Fc. The broad irreversible oxi-dation at ~0.6 V vs. Fc$^+$/Fc is assigned to the oxidation of the chloride anions[33,36]. The two components of this process might be associated with a compound-dependent diffusion of Cl$^−$ conterions[33] to the electrode surface[37].

**Optical properties of the complexes**. Figure 4 depicts the absorption and emission data recorded for the complexes under investigation. All four complexes feature very similar absorption

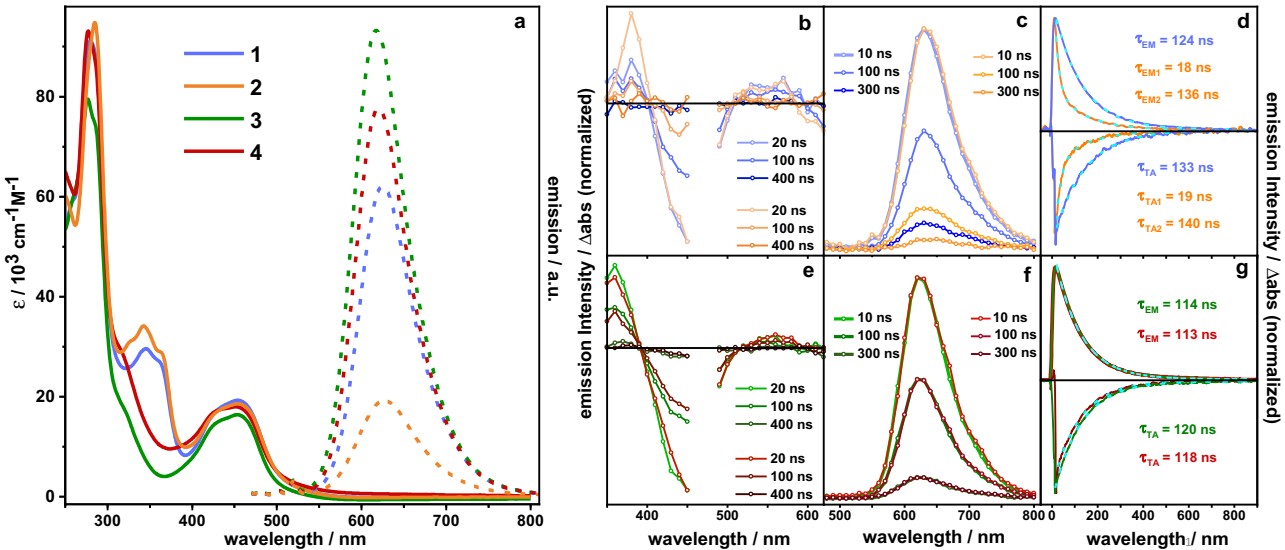

**Fig. 4 Optical properties of the complexes 1, 2, 3 and 4. a** UV/vis absorption (solid) and emission (dashed) spectra ($\lambda_{exc} = 450$ nm) of the samples. **b**, **e** Nanosecond transient absorption spectra of **1** (blue), **2** (orange), **3** (green) and **4** (red) collected upon excitation at 470 nm in acetonitrile. **c**, **f** Time-resolved emission spectra of **1** (blue), **2** (orange), **3** (green) and **4** (red) (experimental data) in acetonitrile excited at 470 nm. **d**, **g** Emission decay profiles of the transient species and normalized transient absorption kinetic traces recorded in the region of the ground-state bleach at 450 nm and the corresponding fit (cyan, dashed). The respectively fitted emission and transient absorption lifetimes are indicated in the plots. For the color codes, see **c** and **f**.

properties characteristic for Ru polypyridine complexes. In acetonitrile, MLCT absorption bands with their maxima in the range of 452 to 454 nm are observed, regardless of the substituent in 5-position of the phenanthroline. Even the further expansion of the π-system in case of complexes **1** and **2** via the alkyne bond does not cause a significant change of the MLCT absorption band which is reminiscent of similar complexes with free alkyne substituents[27,28]. For the alkyne-based complexes **1** and **2**, a broad absorption band at 345 nm is visible. Due to its absence in complexes **3** and **4**, we can assign this to a ligand centered π-π* transition of the delocalized alkyne-bridged phenanthrolines[29]. Additionally, a typical strong absorption band at 280 nm is visible for all complexes caused by π-π* transitions on the tbbpy and phen ligands. All complexes show phosphorescence centered at ~620 nm as typically observed for [(bpy)₂Ru(phen)]²⁺-derived complexes (bpy = 2,2′-bipyridine)[38–41]. Despite the very similar shape of the emission spectra, the emission quantum yield of **1** and **2** is reduced compared to their triazole-containing counterparts. Very intriguingly, comparing the emission quantum yields for the mononuclear ruthenium complexes with the corresponding heterodinuclear ruthenium-rhodium complexes, significant differences become visible (see also Supplementary Table 1). Upon introduction of the Rh^III center, the emission quantum yield of **2** (0.8%) is reduced by more than 90% compared to the mononuclear complex **1** (9.2%; acetonitrile, Ar-atmosphere). In contrast, the emission quantum yield for **3** (15.5%) does only drop by approximately a quarter upon introduction of the Rh^III center (**4**; 11.5%) under identical conditions. Similar behavior of emission quenching upon introduction of a second metal center has been shown in the past[5,32,42,43]. Significant loss of emission intensity has been associated with a high likelihood of electron transfer to the catalytic center[42,44–48]. This behavior is not mirrored by the lifetime of the emissive state where for the alkynyl-bridged systems slightly larger lifetimes (124 ns for **1** and 136 ns (slow lifetime component) for **2**) compared to the triazole-containing complexes (114 ns for **3** and 113 ns for **4**) could be observed. Together with the higher emission intensities

observed for **3** and **4** this indicates that the radiative rate in the alkynyl-bridged systems is lower than in the triazole-linked complexes. We ascribe this finding to a more delocalized long-lived ³MLCT state in the alkynyl-bridged systems. Such electronic delocalization lowers the LUMO and causes the minute bathochromic shift of the emission spectra of **1** and **2** when compared to **3** and **4**. However, the electronic delocalization also partially removes electronic density from the nitrogens coordinating the Ru^III ion and hence reduces the oscillator strength of the ³MLCT→S₀ transition. The bi-exponential emission decay for the binuclear alkynyl complex **2** indicates the coexistence of two distinct ensembles of the excited alkynyl complex (Fig. 4d). We ascribe these ensembles to two conformers of **2** which strongly differ in the electronic coupling between the two metal centers. The observed fast decay time of 18 ns in one of these molecular ensembles indicates quenching of the emissive state by electron transfer to the Rh sphere, while the lifetime of the other species is not changed.

The fact that such fast decay component is not observed in **4** indicates that the triazole bridge, as opposed to the alkynyl bridge, efficiently blocks electron transfer toward the Rh center.

**Catalysis**. To evaluate the photocatalytic performance of the novel heterodinuclear catalysts **2** and **4** together with the benchmark catalyst **5**, first the principal activity of the catalytic center toward NAD⁺ reduction was considered. This was accomplished by running light-independent thermal catalysis with sodium formate as reducing agent[18], which allowed to derive the effect of different BLs on the efficiency of the structurally similar catalytic centers for nicotinamide reduction.

Thermal, formate-driven NAD⁺ reduction by **2**, **4** and **5** was performed at 25–50 °C[18,25]. The effect of different temperatures on the catalytic activity was screened in intervals of 5 °C (Supplementary Fig. 10). All three complexes performed equally over the investigated temperature range and showed temperature dependence (Fig. 5) in accordance with an Arrhenius plot (Supplementary Fig. 11). The average TOF during catalysis

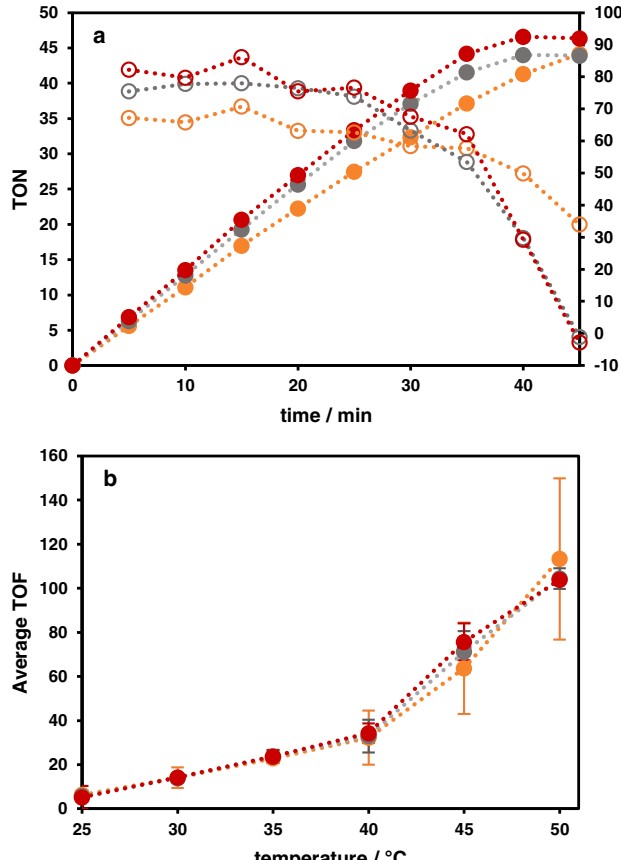

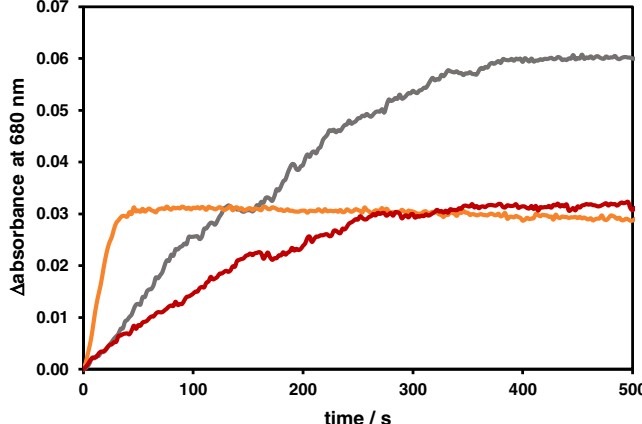

**Fig. 6 Photochemical reduction of the heterodinuclear complexes.** Absorbance at 680 nm during irradiation (470 nm, 54 mW/cm$^2$) of deaerated photocatalyst solutions (0.12 M TEA, acetonitrile/H$_2$O (1/9, v/v)). Complexes **2**, **4** and **5** are represented by orange, red and gray solid lines, respectively.

**Fig. 5 Formate-driven nicotinamide reduction. a** TONs (solid dots, dotted lines to guide the eye) and TOFs (hollow dots, dotted lines to guide the eye) during formate-driven catalysis at 45 °C of complexes **2** (orange), **4** (red), and **5** (gray) (H$_2$O/acetonitrile (9/1, v/v), 50 mM NaHCO$_2$, 5 μM catalyst, 250 μM NAD$^+$). **b** Average TOF (within 90 min or until substrate limitation) during formate-driven catalysis of complexes **2** (orange), **4** (red), and **5** (gray) at temperatures ranging from 25 to 50 °C (dotted lines to guide the eye; error bars represent standard deviation, $n = 2$ independent measurements).

increased 19-fold from 25 to 50 °C for all complexes. Thus, the catalytic activity of the rhodium center is determined by the phenanthroline coordination, while the actual structure of the core unit of the investigated modular BLs (alkyne, triazole or phenazine) does not influence the turnover at the Rh center. This is in line with the very similar redox potentials of the Rh$^{III}$/Rh$^I$ couples (see Supplementary Table 2) of the dinuclear complexes which have been shown to influence the catalytic activity[49]. Furthermore, this finding of identical light-independent turnover at the catalytic centers allows to assign the different photocatalytic activities of the three dinuclear systems (vide infra) to the distinct design of the BLs.

As depicted in Fig. 1, the first step in the photocatalytic mechanism of NADH formation is the light-driven two-fold reduction of the Rh$^{III}$ center to the catalytically competent Rh$^I$ state. This process can easily be monitored with UV/vis spectroscopy due to a broad absorption band centered at ~680 nm originating from a Rh$\rightarrow$phen MLCT transition[14]. An in situ monitoring of the absorbance at 680 nm during irradiation thus allowed us to study the concentration of the Rh$^I$ species over time (Fig. 6 and Supplementary Fig. 12). In the initial 20 s the rate of Rh$^I$ formation was up to 6 times higher for **2** compared to **4** and **5**. Thus, by changing the molecular structure of the BL, significant differences in the formation of

the catalytically competent two-fold reduced Rh$^I$ intermediate can be observed. Furthermore, concentration-dependent measurements were performed using solutions of 2, 5 and 10 μM of **2**, **4** and **5**, respectively (Supplementary Figs. S13 and S14). As for none of the complexes a decrease in the build-up of the Rh$^I$ species upon dilution of the samples was observed, for all complexes sequential intramolecular electron transfers are assumed. Compared to other dinuclear systems[50], the storage of two electrons occurs with relative ease, which possibly results from the well-tuned redox potentials of all subunits or a BL architecture guaranteeing suitable decoupling of chromophore and catalyst subunit in its mono-reduced state. Luminescence quenching studies furthermore indicated that for all three heterodinuclear complexes reduction of Rh$^{III}$ to Rh$^{II}$ proceeded via an oxidative quenching pathway (Supplementary Fig. 15). As the redox potentials for the corresponding Rh$^{III}$/Rh$^I$ couples as well as the Ru$^{III}$/Ru$^{II}$ couples only differed among all three photocatalysts **2**, **4** and **5** on maximum by 70 and 60 mV, respectively (see Supplementary Table 2), the observed effect of the strongly varying rate of Rh$^I$ formation can clearly be associated with the molecular structure of the BL. In **5** accumulation of Rh$^I$ intermediates is followed by photochemical reduction of the tpphz ligand, which further adds to the increasing absorbance at 680 nm at longer times[14]. However, the initial rate of Rh$^I$ formation in **5** is comparable to that of **4**. Under the investigated conditions that resemble the ones used for photocatalytic experiments in terms of donor concentration, initially ~3.5, 0.5 and 0.6% of the photocatalysts **2**, **4** and **5** are doubly reduced per second which would allow ~130, 18 and 22 turnovers per hour as long as the photochemical reduction of the catalyst represents the rate-determining step of the photocatalytic NADH formation.

Above it was shown that all three dinuclear complexes did show very similar behavior in the formate-driven, thermal NAD$^+$ reduction. Furthermore, having established that the photochemical formation of the catalytically competent Rh$^I$ state is feasible in all three heterodinuclear complexes with very different kinetics, we investigated their potential in light-driven catalysis.

For the light-independent (thermal) NADH formation, β-hydride elimination from metal bound formate has been shown to represent the rate-limiting step (Supplementary Fig. 9)[18]. Following our initially raised hypothesis that such energy demanding steps limiting the activity of thermal catalysis can be

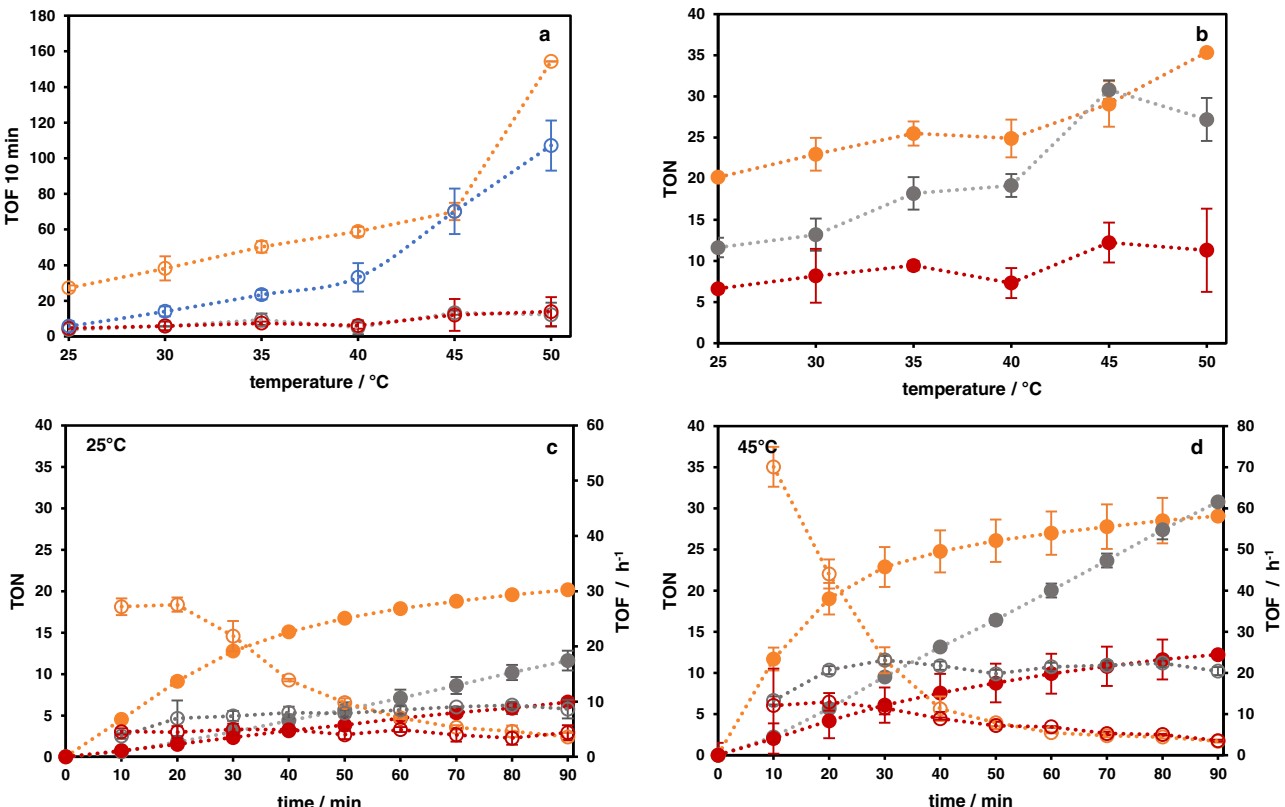

**Fig. 7 Photocatalysis by the heterodinuclear complexes.** Temperature dependent photocatalysis of complexes **2** (orange), **4** (red) and **5** (gray). TON as solid dots, TOF as hollow dots (dotted lines guide the eye; error bars represent standard deviation, $n = 2$ individual measurements for all panels). **a** Temperature dependent TOF within the first 10 min of photocatalysis and average TOF of **2**, **4** and **5** (within 90 min or until substrate limitation) in formate-driven catalysis (blue) for comparison. **b** TON after 90 min of photocatalysis for the three complexes (acetonitrile/$H_2O$ (1/2, v/v), 0.12 M TEA, 0.1 M $NaH_2PO_4$, 5 µM catalyst, 250 µM $NAD^+$). **c** Photocatalysis at 25 °C. **d** Photocatalysis at 45 °C.

circumvented using photocatalysis, it was speculated that the alkyne-bridged system **2** might outpace the thermal catalytic activity for NADH formation. Nevertheless, also for the final hydride transfer from the reductively activated Rh center to the nicotinamide substrate, a non-negligible kinetic barrier exists[51]. Thus, albeit following a different mechanism (see Fig. 1 and Supplementary Fig. 16) avoiding the energy demanding β-hydride elimination step, also photocatalysis was anticipated to display temperature dependence. Consequently, photocatalytic NADH formation was screened between 25 and 50 °C as well. However, a large effect of thermal acceleration on the photocatalytic reaction was only expected if the necessary two-fold photochemical reduction of the catalyst is significantly faster than catalytic turnover at the Rh center itself. Otherwise photochemical two-fold reductive activation of the Rh center would represent the bottleneck of the overall photocatalytic process.

As shown in Fig. 7a, of the three photocatalysts **2**, **4** and **5** the alkyne moiety containing system **2** exhibited the highest initial TOF which also led for most investigated temperatures to the highest TON after 90 min (see Fig. 7b). To ensure that photocatalysis proceeded at all temperatures with the same regioselectivity for NADH formation, emission spectroscopy was employed to monitor the purity of the product. For all three dinuclear complexes the NADH formation selectivity was close to 100% (see Supplementary Table 3 and Supplementary Figs. 19 and 20). Moreover, a strong temperature dependence on the photocatalytic reaction rate was observed for **2**. Between 25 and 50 °C, the initial reaction rate was increased 6.5-fold, whereas for the other heterodinuclear complexes **4** and **5** this increase was only 2.5-fold. However, between 25 and 45 °C, the

increase of the initial photocatalytic reaction rate for all three complexes was ~2.5-fold.

Interestingly, photocatalyst **2** is the only catalyst which exhibited a faster initial catalytic NADH formation in the photocatalytic approach compared to the formate-driven $NAD^+$ reduction (see Fig. 5b). This is opposite to catalysts **4** and **5** which performed worse under photocatalytic conditions compared to the formate-dependent process.

In order to explain why the vastly different initial TOF values of the various catalysts did not affect the TON after 90 min to an equal degree, we performed detailed catalytic measurements at different temperatures, see Fig. 7c, d. Whereas the tpphz-bridged catalyst **5** produced NADH in a nearly constant rate, novel alkyne- and triazole-bridged photocatalysts **2** and **4** exhibited significantly decreasing photocatalytic output with progressing irradiation time. In addition, as **2** reached substrate limitation at 50 °C already after 20 min, the substrate concentration was increased by a factor of 4. However, this only led to a rise in attainable TONs to 60, i.e., by only 42% (Supplementary Fig. 18).

In order to identify dominant deactivation pathways of catalyst **2** under photocatalytic conditions the alkynyl vibration was monitored in resonance Raman experiments (1) first during irradiation with 405 nm (0.12 M TEA, acetonitrile/$H_2O$ (1/2, v/v)) (Supplementary Fig. 22a) and (2) second in presence of a proton source (TFA) during electrochemical reduction of the catalytic center (Supplementary Fig. 22b). The substantial loss of the alkynyl band intensity at 2200 cm$^{-1}$ upon irradiation of **2** under catalytic conditions and during electrochemical reduction in presence of protons proves the loss of the activity boosting alkynyl functionality as one important deactivation pathway.

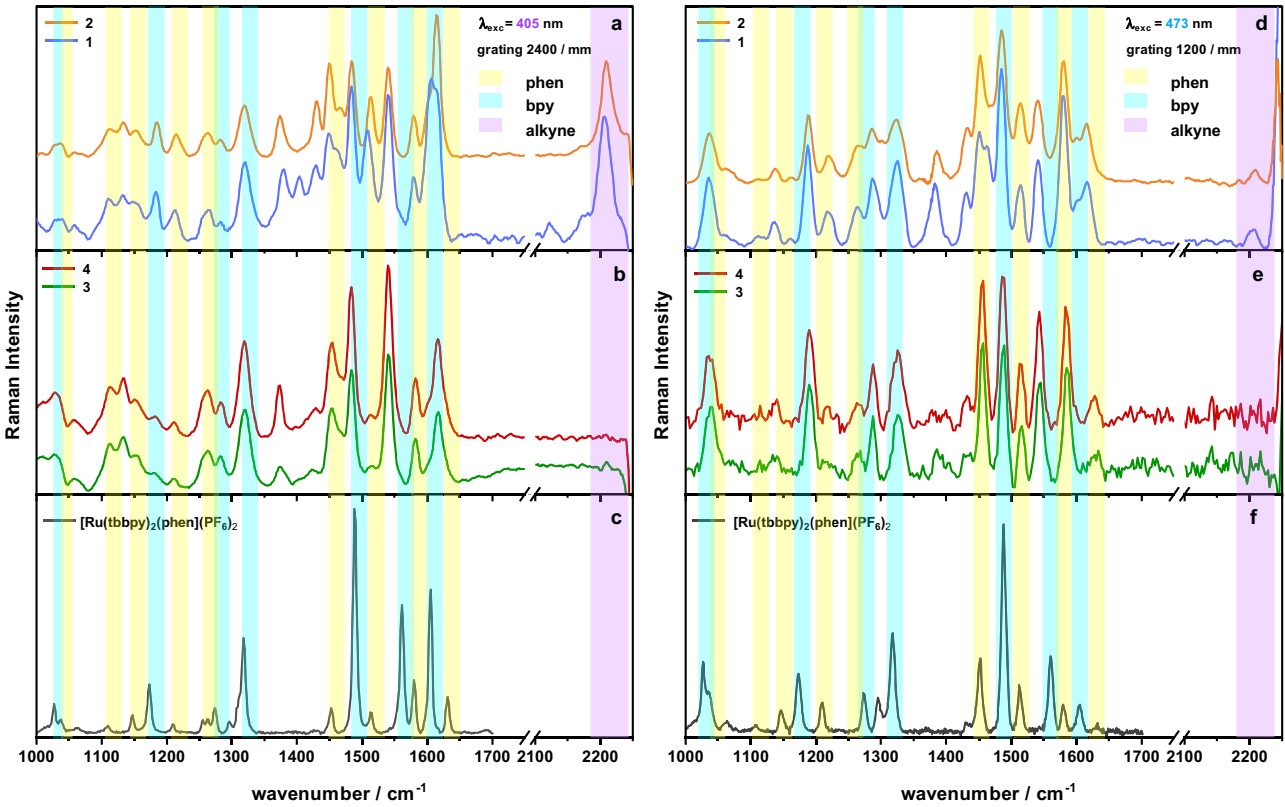

**Fig. 8 Resonance Raman spectroscopy of complexes 1–4 and references.** RR spectra of **1** and **2** in acetonitrile upon excitation at 405 nm (**a**) and 473 nm (**d**). RR spectra of **3** and **4** in acetonitrile excited at 405 nm (**b**) and 473 nm (**e**). RR spectrum of [(tbbpy)$_2$Ru(phen)]$^{2+}$, excited at 405 nm (**c**) and 473 nm (**f**) for comparison. Characteristic Raman modes assigned to bipyridine, phenanthroline and alkyne are highlighted in blue, yellow and purple, respectively.

From experimental design of the resonance Raman experiments it is thus concluded that hydrogenation of the alkynyl bridge occurs. This may take place either by a hydride donating Rh species[19] or by consecutive reactions induced by an excess of electrons stored on the BL upon photoexcitation and in the presence of protons. To corroborate these findings by additional experimental evidence, we performed several experiments. Under formate-driven catalysis conditions but in the absence of NAD$^+$, the π-π* band which we assigned to the conjugated BL of complex **2**, degrades within 60 min (Supplementary Fig. 23a), suggesting loss of the interconnecting alkyne functionality by a hydride donating Rh species. This is also found in an intermolecular approach for the mononuclear complex **1** if [(phen)Rh(Cp*)Cl]Cl is used as hydrogenation catalyst under similar conditions (Supplementary Fig. 23b). The same behavior, although significantly faster can be found for **2** upon irradiation under photocatalytic conditions but in the absence of NAD$^+$ (Supplementary Fig. 23c). Subsequent addition of nicotinamide to this solution after photochemical loss of the alkyne functionality led to a catalytic performance similar to that of late stages of catalysis for complex **2** (Fig. 7 and Supplementary Fig. 23d), thus verifying the necessity of the intact alkyne linker for high photocatalytic activity.

**Photoinduced charge transfer dynamics**. To corroborate the catalytic evaluation of the catalysts with mechanistic information on the intramolecular light-driven charge transfer we will now consider resonance Raman and transient absorption spectroscopy[14].

Resonance Raman (rR) spectra reveal the structure of the Franck-Condon point of absorption and as such the starting geometry for electron transfer in the electronically excited state. rR spectra of **1–4** have been recorded at 405 nm and 473 nm excitation, i.e., in resonance with the $^1$MLCT transition, and reveal characteristic differences between the complexes bearing the alkyne and the triazole linkage. This is exemplified in Fig. 8, which compares the rR spectra with the spectrum of [(tbbpy)2Ru(phen)](PF6)2 as reference.

First, the spectra of triazole moiety containing **3** and **4** are very similar irrespective of the excitation wavelengths (Fig. 8b, e). In addition, the spectra closely resemble the spectrum of the reference compound [(tbbpy)$_2$Ru(phen)]$^{2+}$ (compare Fig. 8b, c, e, f)[40,52–55]. This indicates that the localization of the Franck-Condon region and spatial extension of the $^1$MLCT state within the complexes remains unchanged upon introducing the [(phen)Rh(Cp*)Cl]-moiety (or the phen-fragment only) via Click-chemistry. The situation is different for **1** and **2**, for which the Raman spectra are not only distinct from each other, e.g., at around 1400, 1450 and 1600 cm$^{-1}$, but also differ from [(tbbpy)$_2$Ru(phen)]$^{2+}$. Specifically, the relative intensity ratio of the Raman bands is different, e.g., the strongest band of [(tbbpy)$_2$Ru(phen)]$^{2+}$ at 1490 cm$^{-1}$ at 405 nm excitation is weaker than the bands at 1550 and 1600 cm$^{-1}$ in the alkyne-linked complexes. This indicates, that the $^1$MLCT of the alkyne-linked complexes differs from the $^1$MLCT of [(tbbpy)2Ru(phen)]$^{2+}$. The -C≡C- triple bond stretching vibration is visible as a prominent band in the resonance Raman spectrum both in **1** and in **2** at 405 nm and as weak band at 473 nm excitation. This indicates that the -C≡C- motif allows for interaction of the two phen spheres in the $^1$MLCT localized on the BL. We speculate that the $^1$MLCT is delocalized across the -C≡C- bridge, and hence has distinctly different spectral features than in [(tbbpy)$_2$Ru(phen)]$^{2+}$. This is in line with results on the first reduction of **1**, i.e., reduction of the BL, in a spectro-electrochemical resonance Raman experiment[14,55–62]. Here the

vibrational mode associated with the -C≡C- triple bond is shifted to lower wavenumbers upon reduction (Supplementary Fig. 24). This further corroborates that extension of the phen-ligand by the alkynyl functionality extends the chromophoric system of the ligand thus enabling it to function as an electron acceptor in a MLCT transition even if reduced once by electrochemical means[57]. The downshift of the respective vibrational mode upon reduction of **1** reflects the increased electron density across the -C≡C- triple bond upon electrochemical reduction of the complex. Another spectral feature is the band at $1400\,cm^{-1}$, which is only visible in **1** upon 405 nm excitation. We associate the band with a C-N vibration of uncoordinated phenanthroline ligand. Upon complexation with the [Rh(Cp*)Cl]-center the band disappears. The absence of the $1400\,cm^{-1}$ band in the rR spectrum of **1** recorded upon 473 nm excitation shows a less extended $^1$MLCT upon red-shifting the excitation wavelength.

To investigate differences not only in the initially populated $^1$MLCT state but also in the subsequent intramolecular relaxation, femtosecond transient absorption spectroscopy was performed in acetonitrile and dichloromethane upon excitation at 400 and at 470 nm, i.e., in the high and low energy flank of the $^1$MLCT absorption band. Reminiscent of the findings from the rR data, triazole bridged **3** and **4** reveal an excited state dynamics, which is very much alike the dynamics observed for $[(bpy)_2Ru(phen)]^{2+}$. The corresponding data are summarized in Supplementary Figs. 25–28.

Irrespective of the [Rh(Cp*)Cl]-center being present or not, the ground-state bleach, reflecting the $^1$MLCT absorption band, is accompanied by a broad and unstructured excited state absorption at wavelengths longer than ~500 nm, which stems from ligand-to-metal charge transfer transitions, characteristic for $^3$MLCT states in $[(bpy)_2Ru(phen)]^{2+}$ complexes[41,63,64]. The more intense excited state absorption at ~360 nm is due to $\pi\pi^*$ transitions of the reduced phen-sphere—also characteristic for $[(bpy)_2Ru(phen)]^{2+}$ complexes[41,65]. The transient absorption data – irrespective of solvent and excitation wavelength – show that excited state relaxation in **3** and **4** does not involve the peripheral phen motif or the [(phen)Rh(Cp*)Cl]-center, respectively. Thus, the data line up with the emission data (see Fig. 4), which show comparably high emission of **3** and **4**, barely influenced by the introduction of the [Rh(Cp*)Cl]-center.

The transient absorption data of **1** and **2** are shown in Figs. 9 and 10. Compared to the spectra of **3** and **4**, the pronounced transient absorption feature at 360 nm, characteristic for the $\pi\pi^*$ absorption of the reduced phen, is absent. Instead, a broad and flat excited state absorption is visible on the short-wavelength side of the ground-state bleach, indicating a more delocalized excess charge density in the long-lived $^3$MLCT compared to $[(bpy)_2Ru(phen)]^{2+}$, **3** or **4**. The formation of the long-lived $^3$MLCT state is reflected in the transient absorption kinetics. Upon excitation of the $^1$MLCT transition at either 400 or 470 nm in **1** and **2**, a kinetic component is visible, which is characterized by a positive $\Delta OD$ signature in the range of the ground-state bleach. For **1** and **2** dissolved in dichloromethane the decay of this component is characterized by a characteristic time constant of ~250 ps (**1**) and ~220 ps (**2**, see Supplementary Tables 4 and 5 for a summary of the characteristic time constants describing the kinetic components underlying the sub-ns processes in the complexes). For dichloromethane as solvent the respective component has a large spectral contribution and a characteristic maximum at about 605 nm. This component is atypical for $[(bpy)_2Ru(phen)]^{2+}$ complexes and is assigned to electronic relaxation on the alkynyl-extended phen-ligand. Spectrally, the component is associated with a decay of the phen•-associated ESA at 360 nm and a build-up of an excited state absorption on the short-wavelength flank of the accessible probe window. For the mononuclear species **1** the long-lived

differential absorption spectrum shows stronger contributions below 360 nm than the dinuclear species **2**. However, in both complexes this resulting "plateau-like structure" of the $\Delta OD$ spectra below ~400 nm is distinct from the isolated differential absorption peak at 360 nm, which is observed for **3** and **4**. In addition, the 220 ps/250 ps component causes a red-shift of the ESA in the visible, which also becomes increasingly flat and unstructured during the underlying process. Keeping in mind the rR spectra indicating excitation of a $^1$MLCT state, which is already initially delocalized involving the -C≡C- bond, and the reduced emission yield of **1** and **2**, we assign the 220 ps/250 ps processes to charge density relaxation on the BL. This charge density relaxation redistributes the electron density over the two phen spheres and as a consequence a long-lived $^3$MLCT state is formed, which shows altered ligand-associated $\pi\pi^*$ absorption. The different spectral contributions at below 360 nm, when comparing **1** and **2**, might indicate an enhanced charge localization on the phenanthroline of the [(phen)RhCp*Cl] fragment upon Rh coordination.

Also, for **1** dissolved in the more polar solvent acetonitrile the "plateau-like structure" of the $\Delta OD$ spectra below ~400 nm is visible at long delay times, indicating that a similar long-lived state is populated as in dichloromethane. However, the characteristic kinetic processes yielding the build-up of this long-lived state are different. A sub-10 ps component (the actual value depending on the pump wavelength, for comparison see Supplementary Figs. 29 and 30) jointly with a 280 ps reflect (1) a decay of an ESA in the range of the GSB, reminiscent of the spectral features causing the red-shift of the ESA in the visible spectral range for **1** in dichloromethane and (2) the decay of the phen•-associated ESA at 360 nm and a build-up of ESA on the short-wavelength flank of the accessible probe wavelength. For **2** dissolved in acetonitrile the data indicates only minor ground-state recovery and the loss of the ESA peak at ~570 nm, yielding a long-lived species, whose UV-ESA reveals a localized phen• radical. Taken together with the rR data indicating an initially excited state being delocalized over the -C≡C- band of the BL and the minute emission intensity of **2** we suggest that the phen• radical is centered on the [(phen)RhCp*Cl] sphere. The localization of the excess electron density of the long-lived $^3$MLCT state in **2** on the [(phen)Rh(Cp*)Cl] fragment of the complex is assisted by the high polarity of the solvent favoring the population of electronic states with increased dipole moments.

Considering the spectroscopic characterization, we would like to point out that for **1** and **2** the excited state electron density already at the Franck-Condon point of absorption is (partially) extended over the alkynyl bond. In contrast, the rR spectra of **3** and **4** strongly resemble the spectrum of the reference compound $[(tbbpy)_2Ru(phen)]^{2+}$, i.e., pointing to the access electron density being localized on the phen-sphere coordinating the Ru ion at the Franck-Condon point. Also, the emission properties of **3** and **4** are essentially similar, while coordination of the Rh center to **1** (yielding **2**) yields to a fast 18 ns emission deactivation component. We relate this additional emission decay channel to electron transfer from the emissive $^3$MLCT$_{phen}$, with its excess electron density localized on the phen-sphere coordinating the Ru, to the coordination sphere of the Rh center—an electron transfer that is apparently inhibited in the presence of the triazole-bridge. Judging by the same lifetimes as obtained from time-resolved emission and ns time-resolved transient absorption experiments (Fig. 4), the charge separated state is short lived, such that its decay is hidden beneath its 18 ns formation kinetics.

The pivotal role of the bridging architecture is also reflected in the sub-ns transient absorption kinetics. This kinetics show a

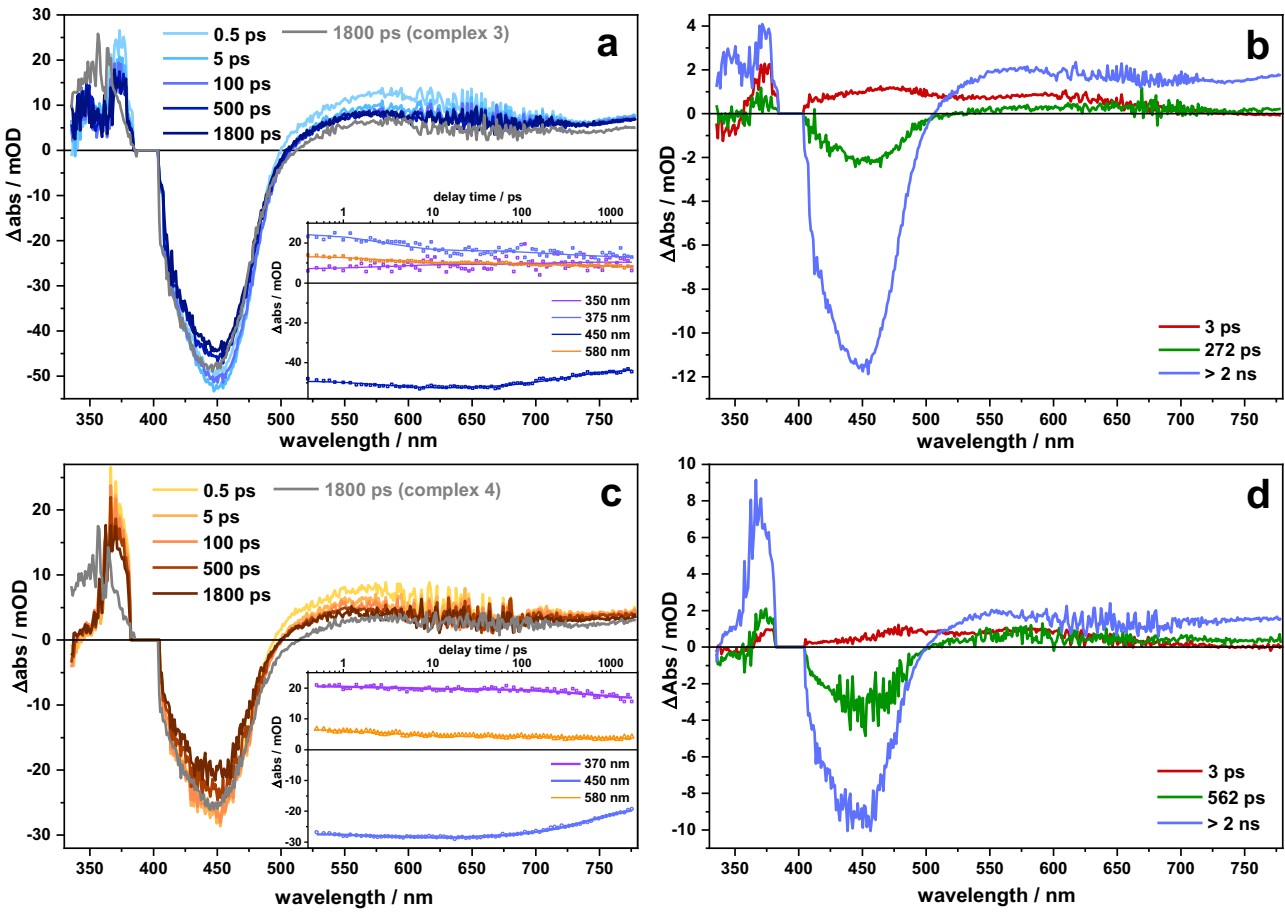

**Fig. 9 Transient absorption spectroscopy in acetonitrile.** Transient absorption spectra recorded for **1** (**a**) and **2** (**c**) in acetonitrile upon pumping at 400 nm at different delay times. Kinetic traces of **1** (Inset in **a**) and **2** (Inset in **c**) at selected wavelength. Decay-associated spectra and corresponding time constants of **1** (**b**) and **2** (**d**) derived from a global multiexponential fit applied on the transient absorption data.

charge density shift within the BL of **1** and **2** (220 ps/250 ps component, respectively) likely associated with structural relaxation of the alkynyl bridge. As inferred from the excited state absorption band below 400 nm, this charge density shift leads to a partial electron transfer from the phenanthroline sphere coordinating the Ru ion to the second coordination sphere. This charge density shift diminishes the contribution of Ru-$^3$MLCT$_{phen}$ to the long-lived excited state and consequently reduces the emission quantum yields of **1** and **2** when compared to **3** and **4**.

## Discussion

Inferred from the formate-dependent, thermal catalysis experiments and very similar electrochemical properties of the Rh$^{III}$ center, the fundamental catalytic activity of the various [(phen)Rh(Cp*)Cl] moieties toward NADH formation in the heterodinuclear complexes **2**, **4** and **5** is equal. Thus, the different photocatalytic activity of these complexes can be ascribed to the impact of the altered structure of the BL on the photochemical reduction of the catalyst. The analysis of substantially reduced photocatalytic activity for NAD$^+$ reduction upon light-driven destruction of the alkyne linker in **2** fortified this hypothesis. Investigations on the rate of Rh$^I$ formation showed (Fig. 6), that the transfer of two electrons toward the rhodium center occurred most effectively for complex **2**. For the other photocatalysts **4** and **5** the photochemical formation of the Rh$^I$ state was ~6–7 times slower compared to **2**. This correlates well with the differing initial TOF values obtained in the

photocatalytic process, where **2** performed ~6 times faster than the two other investigated systems. Thus, it is evident that the improved photocatalytic activity of **2** originated from its superior ability to photochemically form the active Rh$^I$ state.

Moreover, the strong increase in the rate of NADH formation of **2** between 45 and 50 °C might indicate that only after this additional thermal input, catalytic turnover at the Rh center is sufficiently fast to keep up with the efficient photochemistry of the system and to release its full potential. This is well-reflected by the results of the photochemical reduction of **2** (Fig. 6) which suggested a TOF of at least ~130 h$^{-1}$ under exclusively photochemistry-limited conditions. Thus, at lower temperatures than 50 °C, the catalytic turnover at the Rh center limits the photocatalytic NADH formation rate of the overall process in catalyst **2**.

It should additionally be noted that the rate of forming the Rh$^I$ state is not directly correlating with the rate of the first electron transfer to the Rh center. Albeit this is true when comparing novel photocatalysts **2** and **4**, the previously reported tpphz-based system **5** does not follow this trend. In the latter photochemical reduction of the Rh$^{III}$ center with the first electron occurs within less than 0.5 ns[14]. In order to explain the observed differences in forming the Rh$^I$ state it is thus clear that other factors need to be considered as well such as charge recombination rates regenerating the initial Ru$^{II}$-Rh$^{III}$ state or electronic communication between the two metal centers in their onefold or two-fold reduced intermediates enabling light-driven discharging of the reduced catalyst to a greater or lesser extent[14].

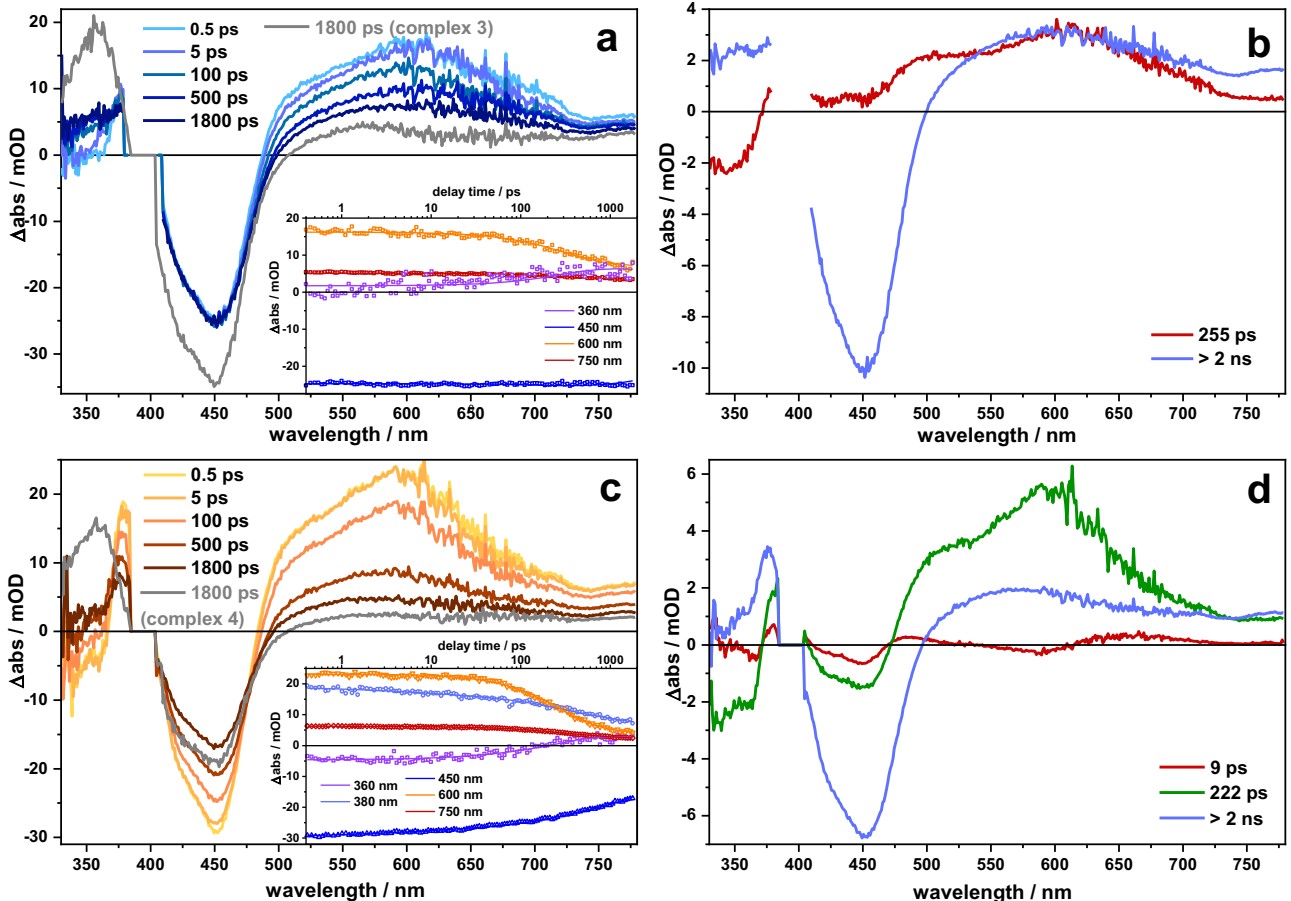

**Fig. 10 Transient absorption spectroscopy in dichloromethane.** Transient absorption spectra recorded for **1** (**a**) and **2** (**c**) in dichloromethane upon pumping at 400 nm at different delay times. Kinetic traces of **1** (Inset in **a**) and **2** (Inset in **c**) at selected wavelength. Decay-associated spectra and corresponding time constants of **1** (**b**) and **2** (**d**) derived from a global multiexponential fit applied on the transient absorption data (**c**).

Lastly, the different time profiles of the photocatalytic NADH formation among the three investigated catalysts indicated that the stabilities of the systems are vastly different. Whereas the tpphz-based catalyst **5** generated NADH with a constant TOF, novel compounds **2** and **4** showed a decreasing activity with increasing irradiation time. Although by increasing substrate availability 4-fold an increase of the overall TON by more than 40% was observed for catalyst **2**, the catalyst deactivates rather rapidly within about 20 min as well. By the various experimental evidence presented above, it is concluded that loss of the alkyne functionality via a hydrogenation process is one important process responsible for the inactivation. Thus, by replacing the tpphz BL with the new alkyne-bridged bis-phenanthroline motif, long-term stability is lost but efficient catalysis for a short period of time is gained.

In summary we presented the two novel heterodinuclear [(tbbpy)$_2$Ru-BL-Rh(Cp*)Cl]$^{3+}$ photocatalysts **2** and **4** and characterized them spectroscopically in detail. Moreover, the heterodinuclear complexes were used for selective formate-driven and photocatalytic formation of biologically relevant NADH and the efficiency of this process was compared to a related tpphz-based benchmark photocatalyst (**5**).

It was shown that the actual molecular structure of the unit connecting the two phenanthroline spheres of the BLs drastically impacts the photophysical properties as well as the photocatalytic activity. Whereas the alkyne linker in **2** allows for intramolecular electron transfers to the Rh center via formation of a $^3$MLCT state involving the triple bond within ~20 ns, the triazole in

contrast behaves insulator-like thus prohibiting similarly efficient reductive activation of the catalyst.

As intended by the design of the complexes, the light-independent, thermal catalytic activity of all three complexes was nearly identical. Thus, these catalysts proved to be ideally suited for a detailed comparison of the impact of the linking strategy within the BL on their photocatalytic activity. Importantly, the rate of photochemical formation of the two-electron reduced Rh$^I$ center could directly be correlated with the photocatalytic activity. For the less active triazole and phenazine linked systems **4** and **5**, sluggish photochemical activation of the catalyst was identified as the major activity limiting step. In contrast, at temperatures up to 45 °C, catalytic turnover at the photochemically activated Rh center in the alkyne-linked system **2** represented its bottleneck. Nevertheless, by circumventing the energy demanding β-hydride elimination of metal bound formate, the light-driven NADH formation by complex **2** was at almost every temperature investigated more efficient than the mechanistically much simpler thermal process. This highlights that by careful design of the molecular structure of supramolecular photocatalysts, photocatalytic processes are able to compete with classical light-independent catalytic processes.

Finally, in combination with the identification of deactivation processes, the easy to analyze separation of photochemical activation and catalytic turnover using [(NN)Rh(Cp*)Cl]-based supramolecular photocatalyst renders them among the best to mechanistically analyze systems. The results at hand clearly showed that for future optimized photocatalytic activity a high rate for two-fold reductive activation of the Rh center—not a high rate for the first electron

transfer—and chemical inertness under catalytic conditions need to be combined.

## Methods

**Sample preparation for photo-driven Rh[I] generation experiments**. For photo-driven Rh[I] generation experiments, 15 nmol (final concentration: 5 μM) of the respective compound were added to a glass vial via a stock solution and the solvent was evaporated. Afterwards, the vial was introduced into a glovebox under argon atmosphere. To the vial, 1000 μl of acetonitrile and 2000 μl of a freshly prepared aqueous TEA solution (final concentration 0.12 M) were added. The content of the vial was mixed, and the solution was then transferred into a cuvette with a 10 mm path length (Starna Scientific). Irradiation was performed using one LED stick emitting blue light ($\lambda = 463$ nm $\pm$ 12 nm, 45–50 mW/cm$^2$) through the top opening of the cuvette inside the glovebox. UV/vis spectra were recorded continuously every 2 s.

**Sample preparation for formate-driven catalysis**. For formate-driven catalysis experiments, 15 nmol of the respective compound (final concentration: 5 μM) was added to a glass vial via a stock solution and the solvent was evaporated. Afterwards, the vial was introduced into a glovebox under argon atmosphere. To the vial, 300 μl of acetonitrile, 2.4 ml of a freshly prepared aqueous NaHCO$_2$ solution (final concentration 50 mM) and 300 μl of a freshly prepared aqueous NAD$^+$ solution (final concentration 250 μM) were added. The content of the vial was mixed, and the solution was then transferred into a cuvette with a 10 mm path length (Starna Scientific) and sealed with a screw cap to exclude oxygen diffusion into the sample. The cuvette was placed in a water bath set to the required temperature. UV/vis spectra were recorded after defined time intervals, and catalytic turnover was determined based on the extinction coefficient (5670 l mol$^{-1}$ cm$^{-1}$) at 345 nm of the formed NADH (nicotinamide adenine dinucleotide).

**Sample preparation for photocatalysis experiments**. For photocatalysis experiments, 15 nmol (final concentration: 5 μM) of the respective compound was added to a glass vial via a stock solution and the solvent was evaporated. Afterwards, the vial was introduced into a glovebox under argon atmosphere. To the vial, 1 ml of acetonitrile, 1.7 ml of a freshly prepared aqueous triethylamine/NaH$_2$PO$_4$ solution (final concentration 0.12 M/0.1 M) and 300 μl of a freshly prepared NAD$^+$ solution (final concentration 250 μM) were added. The content of the vial was mixed, and the solution was then transferred into a cuvette with a 10 mm path length (Starna Scientific) and sealed with a screw cap to exclude oxygen diffusion into the sample. Irradiation was performed using one LED stick emitting blue light ($\lambda = 463$ nm $\pm$ 12 nm, 45–50 mW/cm$^2$) while the cuvette was placed in a water bath set to the required temperature. UV/vis spectra were recorded after defined time intervals of irradiation, and catalytic turnover was determined based on the extinction coefficient (5875 l mol$^{-1}$ cm$^{-1}$) at 345 nm of the formed NADH (nicotinamide adenine dinucleotide).

**Reporting summary**. Further information on research design is available in the Nature Research Reporting Summary linked to this article.

## Data availability

The data generated in this study have been deposited in Zenodo under accession code https://doi.org/10.5281/zenodo.5837780. Synthetic details, information on the experimental setups and additional experimental data generated in this study are provided in the Supplementary Information.

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

## Acknowledgements

We thank the German Science Foundation for funding via the TRR 234 CataLight (project number 364549901; project A1, C.M., B.D.-I. and S.R.; project A4, S.R.), the Studienstiftung des deutschen Volkes (PhD scholarship, P.W.) and the Fonds der Chemischen Industrie (Kekulé-Stipendium, C.M.).

## Author contributions

P.W. synthesized and characterized the molecules, P.W. and A.K.M. performed catalysis experiments, L.Z., C.M. and C.L. performed time-dependent spectroscopy and Raman experiments. P.W., L.Z., A.K.M., B.D. and S.R. wrote the manuscript with help from all other authors.

## Funding

## Competing interests

The authors declare no competing interests.
