## [Peer Review File · Nature Communications]

Reviewers' Comments:

Reviewer #1:

Remarks to the Author:

I have read with interest the manuscript "Which bridge to cross, which mountain to climb – Supramolecular Photocatalysis Outpacing Conventional Catalysis" by Rau, Dietzek and their groups. As for any Rau paper, the technical aspects of the research are superbly developed, the experiments are expertly designed and executed and the amount of methods is breathtaking.

The manuscript deals with an interesting topic, namely the preferred pathway for ET that prompts catalysis, the results seem well interpreted (although I disclose, I am not a photophysics expert), and presented in rich detail.

However, I am not 100% certain whether this is a Nature paper, or if in its present form it is better suited to another journal such as Chem Sci or even ACIE, if substantially shortened. The reason for this rationale is that the study –as technically well performed as it is– does not deliver on the enticing title. The conclusions are general and do not capture the imagination of the reader. Importantly, they do not address explicitly why are photocatalysts outpacing conventional catalysis, as a Nature paper would be expected to do.

The fact that photocatalysts 4 and 5 performed worse than 2 was an unexpected turn of events, as the paper was built (at least to my perception) around species 4. Additionally, species 5 was introduced mid-way in the conversation under "Activity of the Catalyst" and is difficult to visualize for anyone not very familiar with this chemistry.

An interesting but not discussed fact, is that the Rau/Dietzek systems seem to handle subsequent 2e ET with ease. A recent paper published in ACIE early this year has shown the need for two Ru moieties to enable the M(I) state in a 3d metal catalyst. This difference explained would be most welcome by many in this community.

I must say –and emphatically so– that I like the idea of Nature better serving the metal/molecular community by publishing more such papers. I myself have considered the journal a few times, just to send the manuscripts to other top journals that are friendlier to molecular transition metal chemistry. This submission and a recent paper on a Ru(IV) water oxidation intermediate are positive signs of change.

Therefore, I would recommend acceptance based on a substantial revision were the authors rethink the format of the paper to favor a shorter, hypothesis-driven study that connects the bigger picture with the momentous promise of the title.

Assuming the authors would consider these changes, a few minor points in style are offered:

1. Introduction: it seems lengthy and with some redundant information about challenges of the century, etc.; the meaning of the sentence "...bromine moieties in the (paradigm) tpphz..." is not clear; no clear hypothesis can be found, although the sentences including "In order to unequivocally clarify...", and "...the assignment of the bottleneck in photocatalysis difficult...", point out to that direction.
2. Results: are elemental analyses available? I suggest to replace sections named as methods (electrochemistry, absorption, etc.) by the information the methods provide, i.e. "Redox properties" "electronic properties" etc.; What is the origin for the redox processes at -2.5V in 3? The processes attributed to Cl have at least 2, but possibly 3 components suggesting solution equilibria. Please address this in the text.
3. Catalytic activity: please remind the audience of the nature of your catalysis, as often as possible to help with the flow.
4. General: a minor grammar check would remove some very few oddly-constructed sentences.

Reviewer #2:

Remarks to the Author:

In the paper "Which bridge to cross, which mountain to climb – Supramolecular Photocatalysis Outpacing Conventional Catalysis" the authors describe the synthesis and characterization of two

new heterodinuclear photocatalysts. In particular the authors have demonstrated that is fundamental to have the high rate for two-fold electron transfer to the metal centre. I suggest few minor revisions. In the inset of figure 3 the author report that used Ag/AgCl as reference electrode, but in the supporting information have reported Pt wire as reference electrode, which is correct?

In my opinion the emission differences between the compounds will be clearer if discussed in terms of quantum yields and not in terms of intensity. Moreover, all the luminescence lifetime values should be added in the table S1.

Reviewer #3:

Remarks to the Author:

This manuscript describes supramolecular chromophore-catalyst molecules linked through varied bridging ligands while retaining the same coordination properties at the chromophore and catalyst. The catalysts are evaluated for the photochemical reduction of NAD⁺ to NADH, and achieve turnover frequencies exceeding the thermal reaction. Most importantly: the series of molecules provide the authors an opportunity to evaluate photochemical bottle-necks for this two-electron-one-proton reaction. Typically, charge transfers deep into the photochemical cycle are much more difficult to study than the initial steps, and the authors' observations/conclusions represent important findings in this field. Characterizations are thorough and purities are good according to SI data. I recommend reconsideration after minor revisions.

Comments:

-On page 7, the sentence "The emission intensity in dichloromethane of 1 drops by 82% in comparison to 2 upon introduction of the RhIII center" is confusing since 2 is the same as 1 + introduced Rh. The following sentence comparing 3 vs 4 reads correctly.

-On page 8 the biexponential emission decay of 2 is attributed to two channels for emission decay, one including charge transfer. I do not believe this is mathematically the case---two channels for exponential decay should appear as a sum of exponentials, i.e. a single exponential measured decay. A different hypothesis should be explored.

-I find the choice of triethylamine as a quencher interesting, since its quenching rate for the Ru(bpy)₃ excited state is $\sim 10^6$ M⁻¹s⁻¹.

-The section discussing in-situ UV/vis is OK in quantitative terms for comparing formation of Rh(I) across the series. The paper may benefit from nanosecond transient absorption under reductive conditions. These experiments may prove fruitful for observing charge transfer (at least the first one).

-Since there is discussion of the varied nature of the excited states, clearly the chromophores are not "identical" as intended and may therefore exhibit varied excited state potentials and varied rates of reductive quenching. Quenching rates should be provided as verification of the assumption that the second electron transfer is the bottleneck and to prove that it is not an artifact of varied quenching efficiency. If the quenching is the problem, other quenchers should be investigated.

-Resonance Raman monitoring of the alkynyl stretch of 2 to observe the fate of the catalyst was a creative approach. I find it intriguing that the activity of 2 is greatly reduced by hydrogenation as electron transfer across a -(CH₂)-(CH₂)- linkage might be viewed as viable from a design perspective. The authors provided a great analysis on pages 15/16 which discusses the MLCT state delocalized across the bridge and the importance of the alkyne. These observations may be closely related to the dual emission decays of 2 and the decay from two possible non-equilibrated excited states (differing in rotational orientation, maybe?) should be considered.

-On page 16, "the corresponding data is" should be "...data are"

Reviewer 1:

I have read with interest the manuscript “Which bridge to cross, which mountain to climb – Supramolecular Photocatalysis Outpacing Conventional Catalysis” by Rau, Dietzek and their groups. As for any Rau paper, the technical aspects of the research are superbly developed, the experiments are expertly designed and executed and the amount of methods is breathtaking.

The manuscript deals with an interesting topic, namely the preferred pathway for ET that prompts catalysis, the results seem well interpreted (although I disclose, I am not a photophysics expert), and presented in rich detail.

However, I am not 100% certain whether this is a Nature paper, or if in its present form it is better suited to another journal such as Chem Sci or even ACIE, if substantially shortened. The reason for this rationale is that the study —as technically well performed as it is— does not deliver on the

enticing title. The conclusions are general and do not capture the imagination of the reader. Importantly, they do not address explicitly why are photocatalysts outpacing conventional catalysis, as a Nature paper would be expected to do.

Response: In order to address the comment of the reviewer, we i) shortened the text where possible and ii) clarified our conclusions from our experiments where necessary. All changes are highlighted in yellow color in the submitted revised versions of manuscript and supporting information. We hope that our main conclusions are now much easier to grasp. For ease of reading we highlight them here separately:

i) The rate at which a two-fold reduced Rh^I state is formed, not the rate at which the first electron is transferred to the catalyst, can directly be correlated with the rate of photocatalytic NADH formation.

ii) Thus, the alkyne-bridged system, which most effectively forms the Rh^I state, is also the most active photocatalyst.

iii) Thermal catalysis is outpaced by the alkyne-bridged system as the rate-determining step in thermal catalysis (β -hydride elimination of metal-bound formate) is circumvented in photocatalysis.

To address all three points in the abstract, the following has been added:

“With the photocatalytically most efficient alkynyl linked system, it was even possible to overcome the rate of thermal NADH formation by avoiding the rate-determining β -hydride elimination step.”

The fact that photocatalysts 4 and 5 performed worse than 2 was an unexpected turn of events, as the paper was built (at least to my perception) around species 4. Additionally, species 5 was introduced mid-way in the conversation under “Activity of the Catalyst” and is difficult to visualize for anyone not very familiar with this chemistry.

Response: We did not intend to build a story around the dinuclear, triazole-linked photocatalyst 4. The study was rather meant as an unbiased comparison between three photocatalysts differing in the molecular structure of the bridging ligand. We hope that the general adjustment of the text both in the introduction as well as in the results and discussion part helps to overcome this perception. In order to visualize the tpzhz-bridged photocatalyst in the manuscript and to avoid mid-way surprises, species 5 is now included in an updated version of Figure 1 and is introduced within the introduction part of the manuscript as reference systems.

An interesting but not discussed fact, is that the Rau/Dietzek systems seem to handle subsequent 2e ET with ease. A recent paper published in ACIE early this year has shown the need for two Ru moieties to enable the M(I) state in a 3d metal catalyst. This difference explained would be most welcome by many in this community.

Response: After studying the mentioned article (*Angew. Chem. Int. Ed.* **2021**, *60*, 5723–5728) on different Ru-Ni systems (dinuclear Ru-Ni system and trinuclear Ru-Ni-Ru system), our hypothesis why the three dinuclear photocatalysts presented in this study can form the two-fold reduced state with ease is the following: As a consequence of fast Cl loss upon one-fold reduction of the catalyst, [(NN)Rh(Cp*)Cl]-systems show potential inversion of the $Rh^{III/II}$ and $Rh^{II/I}$ couples. This is manifested in an electrochemical two-electron reduction at a comparably low potential of ca. -1.15 V vs. Fc^+/Fc for all three investigated systems. This means, that the thermodynamic driving

force for the second reduction of the catalyst is similar or even slightly higher than the driving force for the first reduction of the catalyst. This is opposite to many other systems where the reductions at the metal become more difficult with each step because the extra charge at the catalyst is not compensated by ligand loss or addition of a proton. Consequently, in these coordinatively more stable systems the driving force for the transfer of the second electron decreases.

However, the mentioned paper indicated that as soon as two electrons are stored in any of the two systems (dinuclear and trinuclear; 1 electron at the Ni-catalyst and 1 electron at a bpy ligand), H₂ evolution can be observed. It is thus tempting to speculate – also with regard to previous literature from our own group – that as long as two electrons can be accumulated in such systems designed for photocatalysis, photocatalytic turnover may be observed under appropriate conditions (pH, concentration of the substrate, etc.). A critical parameter seems to be the storage of the second electron. This process will not only depend on the energetics of the individual components of the system but also on the geometry- and architecture-dependent electronic coupling of those subunits in the one-fold reduced state. It is likely that in the mentioned dinuclear Ru-Ni system, the one-fold reduced Ni center serves as tremendously efficient reductive quencher for the excited Ru state thus precluding extraction of a second electron from the sacrificial donor and thus just triggering the back and forth transfer of electron density. This is no problem if the second electron comes from outside of the system, i.e. by the added [Ru(bpy)₃]²⁺. In the trinuclear Ru-Ni-Ru system this efficient reductive quenching may be somewhat reduced thus allowing the storage of the desperately needed second electron.

We also verified experimentally, that the processes we observed can very likely be ascribed to intramolecular processes. We therefore investigated the concentration dependent Rh^I formation and did not observe a decrease in the rate for any of the three dinuclear complexes **2**, **4** and **5** upon lowering the respective concentration of the photocatalyst from 10 μM to 5 μM and 2 μM. The data is presented in Figure S13.

Taken together, the above given explanation as well as the finding on the “intramolecular nature” of the observed process for the three dinuclear complexes **2**, **4** and **5** is now also included in the manuscript. Considering the suggestions of the reviewer to keep the manuscript as short as possible it reads as follows in the section “Light-driven formation of the Rh^I state”:

*“Furthermore, concentration-dependent measurements were performed using solutions of 2 μM, 5 μM and 10 μM of **2**, **4** and **5**, respectively (see Figure S13-S14). As for none of the complexes a decrease in the build-up of the Rh^I species upon dilution of the samples was observed, for all complexes sequential intramolecular electron transfers are assumed. Compared to other dinuclear systems, the storage of two electrons occurs with relative ease, which possibly results from the well-tuned redox potentials of all subunits or a BL architecture guaranteeing suitable decoupling of chromophore and catalyst in its mono-reduced state.”*

I must say —and emphatically so— that I like the idea of Nature better serving the metal/molecular community by publishing more such papers. I myself have considered the journal a few times, just to send the manuscripts to other top journals that are friendlier to molecular transition metal chemistry. This submission and a recent paper on a Ru(IV) water oxidation intermediate are positive signs of change.

Therefore, I would recommend acceptance based on a substantial revision were the authors rethink the format of the paper to favor a shorter, hypothesis-driven study that connects the bigger picture with the momentous promise of the title.

Response: As discussed above, we tried to shorten the text where possible adapting a more hypothesis-driven story line. Regarding the promise of the title, i.e. that formate-driven NADH formation was surpassed by the photochemistry of the alkynyl-linked RuRh system, we carried out several changes in the results and discussion part and hope that these changes allow the reader to grasp the main results with ease.

Assuming the authors would consider these changes, a few minor points in style are offered:

1. Introduction: it seems lengthy and with some redundant information about challenges of the century, etc.; the meaning of the sentence "...bromine moieties in the (paradigm) tpphz..." is not clear; no clear hypothesis can be found, although the sentences including "In order to unequivocally clarify...", and "...the assignment of the bottleneck in photocatalysis difficult...", point out to that direction.

Response: We tried to improve our introduction as suggested by the reviewer. In order to shorten the introduction we – for example – deleted the sentence "Therefore, renewables such as wind or solar power have to meet a large share of the energy needs of economy and private households in the near future."

In addition, to clarify the statement on the Br-substituted tpphz bridging ligand, we replaced the old sentence by the following one:

"Examples for such rational synthetic approaches, are the attempts to stabilize excited state properties by introducing electron-withdrawing substituents in the BL by Karnahl et al., which however led to a decrease in electron transfer rate and in turn lowered catalytic activity."

Moreover, to clarify the motivation of the study in the introduction and to implement the central research hypothesis, the following section was included:

"Nevertheless, as light-induced electron transfer within supramolecular catalysts can proceed within the ultrafast time domain, such catalysts hold the potential to be more efficient than thermally operating catalysts. Provided suitable BL architectures can be identified that allow rapid multiple charge accumulation at the catalytic center, the photochemical scenario might outpace the ground-state reactivity of the systems which can be hampered by high activation barriers. In order to investigate this enticing hypothesis a set of reference catalysts is required which are capable of performing the same catalytic reaction in a photochemical and thermal setting. Furthermore, the systems have to allow for the separation of the kinetics of light-driven redox activation of the catalytic center from actual catalytic turnover to identify the overall rate limiting step."

2. Results: are elemental analyses available? I suggest to replace sections named as methods (electrochemistry, absorption, etc.) by the information the methods provide, i.e. "Redox properties" "electronic properties" etc.; What is the origin for the redox processes at -2.5V in 3? The processes attributed to Cl have at least 2, but possibly 3 components suggesting solution equilibria. Please address this in the text .

Response: Elemental analyses are not available (and according to the current sample characterization requirements by the nature portfolio only encouraged, no necessity) , but the

compounds have been checked for purity by measuring $^1\text{H-NMR}$ spectra at very high concentrations and high-resolution MALDI mass spectra that fit to the expected species.

As suggested by the reviewer the individual sections were renamed. The following changes have been performed:

- i) "Electrochemistry" was changed to "Redox properties".
- ii) "Absorption and Emission" was changed to "Optical properties of the complexes".
- iii) "Activity of the Catalytic Center" was changed to "Formate-driven nicotinamide reduction."
- iv) "In-situ UV/vis spectroscopy" was changed to "Light-driven formation of the Rh^{I} state".
- v) "Resonance Raman spectroscopy" was changed to "Localization of the initially populated excited states".
- vi) "Femtosecond pump-probe spectroscopy" was changed to "Excited state relaxation pathways".

With respect to the redox process at -2.5 V vs. Fc^+/Fc in **3**, extensive literature research on molecules featuring isolated triazole units did not provide meaningful results. We nevertheless would like to ascribe it to the reduction of a orbital delocalized over the bridging ligand.

As for $[(\text{N,N})\text{Rh}(\text{Cp}^*)\text{Cl}]$ -complexes not having Cl^- counterions no Cl^- oxidation is observed, the differing irreversible waves attributed to the Cl^- oxidation were attributed in the manuscript as follows:

"The two components of this process might be associated with a compound-dependent diffusion of Cl^- counterions to the electrode surface."

3. Catalytic activity: please remind the audience of the nature of your catalysis, as often as possible to help with the flow.

Response: We thank the reviewer for this comment. During re-evaluation of the manuscript we now remind the audience 8 additionally times on the nature of our catalysis and used the phrases "NAD $^+$ reduction" or "NADH formation".

4. General: a minor grammar check would remove some very few oddly-constructed sentences.

Response: We checked the whole manuscript and improved where it seemed necessary.

Reviewer 2:

In the paper "Which bridge to cross, which mountain to climb – Supramolecular Photocatalysis Outpacing Conventional Catalysis" the authors describe the synthesis and characterization of two new heterodinuclear photocatalysts. In particular the authors have demonstrated that is fundamental to have the high rate for two-fold electron transfer to the metal centre. I suggest few minor revisions.

In the inset of figure 3 the author report that used Ag/AgCl as reference electrode, but in the supporting information have reported Pt wire as reference electrode, which is correct?

Response: As now stated correctly, a Ag wire was used as quasi reference electrode and referenced to Fc^+/Fc . The Pt wire served as counter electrode and a glassy carbon electrode as

the working electrode. The information in the manuscript as well as the SI document are now identical.

In my opinion the emission differences between the compounds will be clearer if discussed in terms of quantum yields and not in terms of intensity. Moreover, all the luminescence lifetime values should be added in the table S1.

Response: As suggested, we determined the emission quantum yields for all complexes in acetonitrile (aerated and de-aerated (Ar)). As requested, the values are now implemented in Table S1 – along with the luminescence lifetime values – and the corresponding section in the manuscript now reads as follows:

“Despite the very similar shape of the emission spectra, the emission quantum yield of 1 and 2 is reduced compared to their triazole-containing counterparts. Very intriguingly, comparing the emission quantum yields for the mononuclear ruthenium complexes with the corresponding heterodinuclear ruthenium-rhodium complexes, significant differences become visible (see also Table S1). Upon introduction of the Rh^{III} center, the emission quantum yield of 2 (0.8%) is reduced by more than 90 % compared to the mononuclear complex 1 (9.2%; acetonitrile, Ar-atmosphere). In contrast, in acetonitrile the emission quantum yield for 3 (15.5%) does only drop by approx. a quarter upon introduction of the Rh^{III} center (4; 11.5%) under identical conditions.”

Reviewer 3:

This manuscript describes supramolecular chromophore-catalyst molecules linked through varied bridging ligands while retaining the same coordination properties at the chromophore and catalyst. The catalysts are evaluated for the photochemical reduction of NAD⁺ to NADH, and achieve turnover frequencies exceeding the thermal reaction. Most importantly: the series of molecules provide the authors an opportunity to evaluate photochemical bottle-necks for this two-electron-one-proton reaction. Typically, charge transfers deep into the photochemical cycle are much more difficult to study than the initial steps, and the authors' observations/conclusions represent important findings in this field. Characterizations are thorough and purities are good according to SI data. I recommend reconsideration after minor revisions.

Comments:

On page 7, the sentence “The emission intensity in dichloromethane of 1 drops by 82% in comparison to 2 upon introduction of the Rh^{III} center” is confusing since 2 is the same as 1 + introduced Rh. The following sentence comparing 3 vs 4 reads correctly.

Response: As suggested we clarified this statement. It now reads as follows (note that emission intensity values were replaced by emission quantum yields due to the comment of another reviewer):

“Upon introduction of the Rh^{III} center, the emission quantum yield of 2 (0.8%) is reduced by more than 90 % compared to the mononuclear complex 1 (9.2%; acetonitrile, Ar-atmosphere).”

*In contrast, in acetonitrile the emission quantum yield for **3** (15.5%) does only drop by approx. a quarter upon introduction of the Rh^{III} center (**4**; 11.5%) under identical conditions.”*

On page 8 the biexponential emission decay of **2** is attributed to two channels for emission decay, one including charge transfer. I do not believe this is mathematically the case - two channels for exponential decay should appear as a sum of exponentials, i.e. a single exponential measured decay. A different hypothesis should be explored.

Response: We apologize for this misleading description and we fully agree with the referee. Both the time resolved emission spectroscopy and transient absorption data of the ground state bleach are characterized by a biexponential decay. This biexponential decay is characterized (both for emission and ground state recovery) by two characteristic lifetimes, i.e. 14 and 140 ns. This points to the co-existence of two distinct ensembles of excited molecules. Only one of those species is affected by quenching when introducing the Rh center, because the decay constant of the second species does not change. It is similar to the mononuclear complex. Additional quenching experiments (see Fig. S15) proof, that quenching is indeed due to the Rh fragment.

From our experimental data we can exclude significant contaminations of the sample of the bimetallic complex with the mononuclear complex. Therefore, the presence of impurities cannot account for the presence of the long-lived species.

We rephrased the corresponding section (page 8) in the discussion of the emission data as follows:

*“The bi-exponential emission decay for the binuclear alkynyl complex **2** indicates the coexistence of two distinct ensembles of the excited alkynyl complex. We ascribe these ensembles to two conformers of **2** which strongly differ in the electronic coupling between the two metal centers. The observed fast decay time of 18 ns in one of these molecular ensembles indicates quenching of the emissive state by electron transfer to the Rh sphere, while the lifetime of the other species is not changed.”*

I find the choice of triethylamine as a quencher interesting, since its quenching rate for the Ru(bpy)₃ excited state is $\sim 10^6 \text{ M}^{-1}\text{s}^{-1}$.

Response: As implied by the reviewer, there exist other sacrificial electron donors than simple tertiary amines such as the herein utilized TEA (or TEOA, triethanolamine) which can easier be oxidized than the amines and thus yield higher rates for the reductive quenching. However, we chose TEA as it is non-absorbing in the UV-area relevant for detection of NADH (i.e. at 340 nm). Other electron donors which often give better results than these simple tertiary amines are N-benzyl nicotinamide (BNAH) or 1,3-dimethyl-2-phenyl-2,3-dihydro-1H-benzo[d]imidazole (BIH) and its derivatives. However, BNAH as well as BIH do absorb at 340 nm and thus make it impossible to use those reducing agents to follow and quantify light-driven NADH formation using optical spectroscopy. As will be discussed below, the rate for reductive quenching may not be highly relevant in our systems as we propose that those systems likely operate via oxidative quenching by the coordinated Rh^{III} metal center / extended bridging ligand.

The section discussing in-situ UV/vis is OK in quantitative terms for comparing formation of Rh(I) across the series. The paper may benefit from nanosecond transient absorption under reductive conditions. These experiments may prove fruitful for observing charge transfer (at least the first one).

Response: The referee is correct that it is worthwhile to study the nature of the long-lived state and thus the charge separated state in the photocatalysts **1**, **2**, **3** and **4** by nanosecond transient absorption (ns TA) spectroscopy (at least the first electron transfer). These investigations were part of this study; however, in the original version of the manuscript, we only showed the normalized transient absorption kinetic traces recorded in the region of the ground state bleach at 450 nm (Fig. 4d and 4g). We have now additionally included the ns TA spectra at different delay times in Fig. 4 (Fig 4b and 4e). Nevertheless, it has to be considered that in the ns TA spectra of the photocatalysts the spectral changes induced by the short-lived component are not observable due to the limited signal to noise ratio and the limited spectral sampling interval (10 nm). In summary the ns TA spectra match the features of the of the long-lived component obtained from fs TA spectroscopy.

We also agree with the referee that studying the photodynamics of the second electron transfer from the Ru photocenter to the Rh catalyst, expecting to result in the catalytic active species is of high interest. We will continue to investigate the photoinduced ultrafast dynamics of intermediates (generated electrochemically) of these photocatalysts by ns-TA-SEC. However, this experimental approach of ns-TA-SEC is still under development in our lab and unfortunately there is no data currently available.

Since there is discussion of the varied nature of the excited states, clearly the chromophores are not “identical” as intended and may therefore exhibit varied excited state potentials and varied rates of reductive quenching. Quenching rates should be provided as verification of the assumption that the second electron transfer is the bottleneck and to prove that it is not an artifact of varied quenching efficiency. If the quenching is the problem, other quenchers should be investigated.

Response: Initially we would like to note, that we did not conclusively state that the second electron transfer is the bottleneck; we rather stated, that the rate of Rh^I formation correlates with photocatalytic activity. However, as stated in the discussion part (see below, rephrased to improve its clarity), we did not solely address this to the fate of the second electron:

*“It should additionally be noted that the rate of forming the Rh^I state is not directly correlating with the rate of the first electron transfer to the Rh center. Albeit this is true when comparing novel photocatalysts **2** and **4**, the previously reported *tpphz*-based system **5** does not follow this trend. In the latter photochemical reduction of the Rh^{III} center with the first electron occurs within less than 0.5 ns. In order to explain the observed differences in forming the Rh^I state it is thus clear that other factors need to be considered as well such as charge recombination rates regenerating the initial Ru^{II}-Rh^{III} state or electronic communication between the two metal centers in their onefold or twofold reduced intermediates enabling light-driven discharging of the reduced catalyst to a greater or lesser extent.”*

Nevertheless, we thank the reviewer for this very thoughtful comment. In order to address it, we measured the luminescence quenching of the mononuclear reference complexes **1**, **3** and [(*tbbpy*)₂Ru(*tpphz*)](PF₆)₂ in de-aerated acetonitrile using triethylamine (Fig. S15). The data obtained were compared to the luminescence quenching for complexes **2** and **4**, e.g. if the Rh

center is introduced, i.e. we compared the luminescence quenching of **1** and **3** by TEA with the luminescence of **2** and **4**. The data is presented in the Supporting Information file as Figure S13. Not only was the oxidative quenching (via comparison of mononuclear and dinuclear complexes) for both new complexes **2** and **4** much larger than the reductive quenching of **1** and **3** by TEA but also was the luminescence quenching in absolute numbers – under conditions relevant for photocatalysis (0.1 M TEA and 5 μ M Rh covalently attached to the Ru sphere) – larger for oxidative quenching than for reductive quenching. It should additionally be noted that the effective concentration of the TEA in its base form in the photocatalytic solution is even lower, i.e. ca. 0.02 M, as TEA (0.12 M) is buffered by NaH_2PO_4 (0.1 M), strengthening the hypothesis that oxidative quenching rather than reductive quenching is the dominant mechanism. In addition, luminescence quenching by TEA is larger for $[(\text{tbbpy})_2\text{Ru}(\text{tpphz})](\text{PF}_6)_2$ than for the very efficiently operating alkyne-bridged system **1**, which is opposite to the outcomes observed for photocatalysis.

To sum up, the data at hand (including the fs pump-probe spectroscopy results) suggest, that – at least the first electron – is transferred to the Rh center via oxidative quenching. The following sentence thus has been added to the section “Light-driven formation of the Rh^{I} state”:

“Luminescence quenching studies furthermore indicated that for all three heterodinuclear complexes reduction of Rh^{III} to Rh^{II} proceeded via an oxidative quenching pathway (Figure S15).”

Resonance Raman monitoring of the alkynyl stretch of **2** to observe the fate of the catalyst was a creative approach. I find it intriguing that the activity of **2** is greatly reduced by hydrogenation as electron transfer across a $-(\text{CH}_2)-(\text{CH}_2)-$ linkage might be viewed as viable from a design perspective. The authors provided a great analysis on pages 15/16 which discusses the MLCT state delocalized across the bridge and the importance of the alkyne. These observations may be closely related to the dual emission decays of **2** and the decay from two possible non-equilibrated excited states (differing in rotational orientation, maybe?) should be considered.

Response: We thank the referee for this positive feedback and agree with the hypothesis that two non-equilibrated excited states differing in rotational orientation are present upon photoexcitation of the photocatalyst **2**. This hypothesis can explain the observation of two decay channels observed by time-resolved emission spectroscopy and the kinetic of the ground state bleach as indicated above.

Furthermore, preliminary experiments on the electrochemically reduced photocatalyst **2** point towards a frequency shift of the alkyne band upon reduction of the catalytic Rh center (and excitation of the Ru photocenter), which indicates on the one hand that the ground state geometry (which shapes the rR spectra) changes due to reduction. Therefore, we suspect a reduction-induced geometry change of the complex (possibly planarization that lowers the LUMO of the phenanthroline) that allows this process to occur due to stronger delocalization of the charges. However, these experiments are too preliminary to be fully discussed in this manuscript and will be part of an elaborate follow-up study.

On page 16, “the corresponding data is” should be “...data are”.

Response: We corrected this accordingly.

We are looking forward to your comments.

With kind regards,

Sven Rau

Reviewers' Comments:

Reviewer #1:

Remarks to the Author:

I have read with interest the revised manuscript. The authors addressed most of my pressing concerns.

It is still unusual to me that a paper of this caliber and in a premier journal omits elemental analyses. While NMR can, indeed detect purity of a sample, MS is limited because it may not detect neutral contaminants.

I recommend approval, pending on the editor's position on the lack of a more thorough analysis.

Reviewer #2:

Remarks to the Author:

the authors have answered satisfactorily to all my observations. the papeer is now suitable for publication in Nature Comunication

Reviewer #3:

Remarks to the Author:

The authors have performed an extraordinary effort to address comments of all reviewers, and they have improved the manuscript greatly.

Additionally, their responses have shown that their observations have seeded future studies and I wish them well in their endeavors.